# Metabolite Sensing GPCRs: Promising Therapeutic Targets for Cancer Treatment?

**DOI:** 10.3390/cells9112345

**Published:** 2020-10-23

**Authors:** Jesús Cosín-Roger, Dolores Ortiz-Masia, Maria Dolores Barrachina, Sara Calatayud

**Affiliations:** 1Hospital Dr. Peset, Fundación para la Investigación Sanitaria y Biomédica de la Comunitat Valenciana, FISABIO, 46017 Valencia, Spain; 2Departament of Medicine, Faculty of Medicine, University of Valencia, 46010 Valencia, Spain; M.Dolores.Ortiz@uv.es; 3Departament of Pharmacology and CIBER, Faculty of Medicine, University of Valencia, 46010 Valencia, Spain; dolores.barrachina@uv.es (M.D.B.); sara.calatayud@uv.es (S.C.)

**Keywords:** G-protein-coupled receptor, metabolite sensing GPCR, cancer

## Abstract

G-protein-coupled receptors constitute the most diverse and largest receptor family in the human genome, with approximately 800 different members identified. Given the well-known metabolic alterations in cancer development, we will focus specifically in the 19 G-protein-coupled receptors (GPCRs), which can be selectively activated by metabolites. These metabolite sensing GPCRs control crucial processes, such as cell proliferation, differentiation, migration, and survival after their activation. In the present review, we will describe the main functions of these metabolite sensing GPCRs and shed light on the benefits of their potential use as possible pharmacological targets for cancer treatment.

## 1. Introduction

G-protein-coupled receptors (GPCRs) are characterized by a seven-transmembrane configuration, constitute the largest and most ubiquitous family of plasma membrane receptors, and regulate virtually all known physiological processes in humans [1,2]. This family includes almost one thousand genes that were initially classified on the basis of sequence homology into six classes (A–F), where classes D and E were not found in vertebrates [3]. An alternative classification scheme [4] divides vertebrate GPCRs into five classes, overlapping with the A–F nomenclature [5].

As their name implies, ligand-activated GPCRs function through their interaction with intracellular G proteins, which are heterotrimeric guanine-nucleotide-binding regulatory proteins. G proteins are formed by a combination of α, β, and γ subunits, and are identified by their Gα monomers, which are grouped into four families (Gαs, Gαi, Gαq, and Gα12) and associated to specific signaling effectors that transduce and amplify signals via second messengers [6]. This is the classic ‘GPCR–G protein’ activation mode, but several novel modes of GPCR activation have been discovered, adding significant complexity to this signaling system [7]. The best explored of these alternative pathways is that mediated by β-arrestin. This intracellular protein was initially associated to desensitization of the G protein-mediated signaling, but was later implicated in receptor internalization, and in the activation of multiple intracellular signaling pathways that may coincide with some signals elicited by G proteins, but often with very distinct cellular consequences [2]. Additionally, cumulating evidence suggests that activated GPCRs can signal through the transactivation of tyrosine kinase receptors [7], which may be of particular relevance in cancer cells [8].

Classically, GPCR receptors were identified as targets of particular biological mediators, such as hormones or neurotransmitters. However, the deciphering of the human genome identified hundreds of GPCRs whose endogenous ligands were unknown. These receptors were included in a hotchpotch and were called orphan GPCRs while waiting for their linking to a particular signaling biomolecule. As part of this process, some of these initially orphan receptors have been identified as targets for molecules belonging to the field of metabolism. Thus, intermediary metabolites that were seen as mere pieces of the cellular energy-making machinery, or more recently as intracellular signaling elements [9], are now recognized as extracellular mediators able to modulate physiological functions or affect pathological processes through the activation of GPCRs in an autocrine or paracrine manner.

This new notion appears of relevance in cancer as altered cell metabolism is a characteristic feature of many tumors. Some of these metabolic changes are analogous to those observed in normal proliferating cells, as they are directly associated to the cell division process and the needs of biochemical building blocks and of energy to sustain these biosynthetic pathways. Other changes are the consequences of genetic alterations in cancer cells that determine loss or gain of function in metabolic enzymes and promote the accumulation of specific metabolites. Finally, cancer and stromal cells often modify their metabolic functioning to preserve cell homeostasis in the altered and evolving environment of a tumor [10].

The present review tries to summarize the existing information about all those GPCRs that, at present, are considered as targets for endogenous metabolites in cancer, with the aim of detecting fields of research that may be the basis for the development of new therapies.

## 2. Metabolite-Sensing GPCRs

This group of GPCRs commenced in 2003 with the identification of GPR40 [11], GPR41, and GPR43 [12] as receptors for fatty acids. They were recognized as such and renamed as free fatty acid receptor-1 (FFA1), FFA2, and FFA3, respectively, by the International Union of Pharmacology (IUPHAR) [13]. This initial cluster of metabolite-sensing GPCRs has expanded in the last years by the addition of other fatty acid receptors and receptors responding to hydroxycarboxylic acids, bile acids, amino acids, protons or the Krebs cycle intermediate succinate [14,15], some of which are still considered orphan receptors [16].

We have classified the metabolite-sensing GPCRs according to the presently recognized endogenous agonists, and the complete list of these receptors with their main information is presented in Table 1. Most of these receptors have shown a differential expression, in particular cancers (Table 2), and the following sections, gather all the data regarding their putative role in cancer initiation or progression.

## 3. Fatty Acid Receptors

Fatty acids are carboxylic acids with a long aliphatic chain whose length is used to classify them as short-chain fatty acids (SCFAs, less than 6 carbon atoms), medium-chain fatty acids (MCFAs, 6–12 carbons), and long-chain fatty acids (LCFAs, 12 or more carbons). Short-chain fatty acids (SCFAs) are major products of gut microbial fermentation while long- and medium-chain fatty acids are derived mainly from dietary triglycerides.

### 3.1. GPCRs for Short Chain Fatty Acids (SCFAs)

SCFAs, including propionate, acetate and butyrate, profoundly affect host health and disease. In normal conditions, plasma levels of these metabolites are low, but, overnight fasting or prolonged starvation significantly increase them. The regulation of intracellular histone deacetylases (HDCAs) is a known pathway stimulated by SCFAs but, the activation of G-protein-coupled receptors (GPCRs) has emerged as a new mechanism of action. SCFAs-sensing G protein-coupled receptors include GPR43 (FFA2 receptor), GPR41 (FFA3 receptor), and GPR109A (hydroxycarboxylic acid receptor 2, HCA_2_ receptor), and, depending on the cell type and ligand, these receptors signal via Gα, Gi, Gq, or β-arrestins, or a combination of them [17]. We compile below the reported literature about the relevance of GPR43 and GPR41 in cancer, while the role of GPR109A is analyzed in the section dedicated to hydroxycarboxylic acid receptors. In addition, we summarize in Table 3 the functional studies performed so far, which described the role of both FFA2 and FFA3 receptors in different types of cancers.

#### 3.1.1. GPR43/FFA2 Receptor

FFAR2 (free fatty acid receptor 2) expression is mainly detected in adipose tissue, leukocyte populations and the gastrointestinal tract [18,19]. Among SCFAs, propionate binds FFAR2 with the highest potency, followed by acetate and butyrate and upon binding of these ligands, the receptor signals through both Gαi and Gaq (Table 1).

The biological relevance of FFAR2 in inflammation and carcinogenesis is still a matter of debate. The presence of FFAR2 was first demonstrated in a human breast cancer cell line and its activation by propionate increased intracellular calcium and the phosphorylation of mitogen-activated protein kinase (MAPK) p38 [20]. However, contradictory results were early reported about the FFAR2 expression in gastrointestinal cancer; a first study revealed a higher FFAR2 expression in colorectal and gastric cancers and the overexpression of this receptor in cancer cells potentiated their growth when xenografted in nude mice [21,22]. In contrast, Tang et al. showed a reduced expression of FFAR2 in most colorectal adenocarcinoma tissues, and their corresponding lymph node metastatic adenocarcinomas, compared with normal colon tissue [18]. Further studies reinforced this last observation in human colon cancer, and demonstrated a protective role for FFAR2 in a variety of mouse models of colon carcinogenesis [23,24,25,26]. Finally, it has been recently reported that the loss of FFAR2 promoted the development of colon adenoma and the progression of adenoma to adenocarcinoma in both humans [27]) and mice [28]). The mechanisms involved in cancer silencing of FFAR2 are still unknown, but it has been shown that the receptor undergoes important post-translational modifications [17].

The functional role of this receptor was demonstrated in human colon cancer cell lines in which FFAR2 mRNA expression was almost absent and the restoration of this receptor, and its activation by SCFAs, inhibited cell proliferation and induced apoptosis [18]. Strong evidence suggest that FFAR2 acts a colon tumor suppressor and its deficiency has been associated with an enhanced downstream cAMP–PKA–CREB pathway, which resulted in epigenetically decreased levels of inflammation suppressors [28] or the activation of the Wnt pathway [24].

Taking together, literature suggests that the activation of GRP43 may be of importance in the therapeutics of cancer but, the reduced expression of this receptor associated to numerous tumors may strongly difficult this aim. Of interest, several stimuli including butyrate [29], resistant starch-containing diets [30] and the natural product, protopanaxatriol saponin [31] have been related with both an increased expression of FFAR2 and anti-proliferative effects. The combined administration of these compounds with SCFAs or FFAR2 agonists may result beneficial in cancer therapy. Several studies have already demonstrated that the administration of synthetic FFAR2 agonists reduced the induced cell proliferation of both B cells [32] and glomerular mesangial cells [33].

#### 3.1.2. GPR41/FFA3 Receptor

FFAR3 (free fatty acid receptor 3) is highly expressed in adipose tissue, endocrine cells, enteric neurones and leukocytes [34] but, in contrast to FFAR2, it is not detected in epithelial cells. Both propionate and butyrate, constitute the most potent agonists for this receptor [12], which couples both Gq and Gi family proteins. Activation of this receptor with propionate is involved in the production of leptin and several gastrointestinal hormones.

In a similar manner to that described for FFAR2, the expression of FFAR3 was early detected in a human breast cancer line [20], and contradictory results have been reported about its expression in human cancer and its role in carcinogenesis. An early study demonstrated that the over-expression of FFAR3 in a cell line prevented the anti-proliferative and pro-apoptotic properties of butyrate [35]. In contrast, a role for FFAR3 in the anti-proliferative synergistic effect obtained with propionate and cisplatin on human hepatocellular carcinoma cells has been reported [36] and a recent study reveals that the activation of FFAR3 by SCFAs inhibit bovine epithelial cells proliferation [37]. Of interest, it has been shown that FFAR3 and FFAR2 form a receptor heteromer, which has the ability to induce p38 phosphorylation and cellular differentiation; the activation of this receptor may constitute a novel approach for drug targeting [38]. 

### 3.2. GPCRs for Medium and Long Chain Fatty Acid (MCFA, LCFA)

A wide range of both saturated and unsaturated fatty acids containing from 6 to 22 carbons (MCFAs and LCFAs) have been shown to act as agonists at GPR40 (FFAR1) and GPR120 (FFAR4). Additionally, MCFAs stimulate a structurally-unrelated and still orphan G protein-coupled receptor (GPR84), while some oleic acid derivatives act as agonists of the GPR119 [16]. In the following section of this review, we will describe the role of these medium or long chain sensitive GPCRs in different types of cancer. In addition, we have included the most relevant functional studies, which have reported pro-cancerous or anti-cancerous effects of FFA1, FFA4, GPR84, and GPR119 receptors in several human cancers in Table 3.

#### 3.2.1. GPR40/FFA1 Receptor

FFA1 receptor is present in the gastrointestinal tract, pancreatic β-cells, and brain. It plays a relevant role in the glucose-dependent insulin secretion and is an important molecular target for metabolism control. Increasing evidence shows that FFAR1 also plays a role in tumorigenesis, migration and metastasis [39].

Several studies establish a link between the expression of GPR40 and cancer progression. Initial studies detected GPR40 overexpression in some cases of insulinoma [40]. More recently, Munkarah et al. observed significantly increased levels of FFAR1 in high grade ovarian carcinoma specimens and higher expression in advanced stage disease. This up-regulation of FFAR1 could be induced by fatty acids, as ovarian tumors from mice fed with high fat diet and epithelial ovarian cells treated with these mediators presented higher expression of this receptor. Pharmacological experiments in cells in culture showing that FFAR1 activation promotes ovarian cancer cell proliferation support the contribution of this receptor to tumor growth [41]. A similar situation has been observed in pancreatic cancer, where FFAR1 overexpression was detected in high grade carcinoma tissues and functional studies determined that this receptor mediated the oleic acid induced proliferation and resistance to cytotoxic agents in pancreatic cancer cells in vitro [42]. Finally, in colorectal carcinomas, the expression of FFAR1 was positively associated with blood triglycerides level and with the presence of metastasis, advanced disease and poor prognosis [43]. The protumoral role of FFAR1 is also supported by a series of studies performed in cultured cells. A stimulating effect of this receptor on cell growth and/or motility has been observed in breast [44,45], melanoma [46], and lung [47] cancer cells. However, opposite effects have been observed in cells from fibrosarcoma [48] or in pancreatic cancer cells [49]. Thus, most studies support a protumoral role of FFAR1, although a dual role of this receptor cannot be excluded at present.

#### 3.2.2. GPR120/sFFA4 Receptor

FFAR4/GPR120 shows little homology with the FFAR1 cluster (shares 10% amino acid homology) but does share various ligands with FFAR1. FFAR4, such as FFAR1, binds medium- to long-chain fatty acids, including omega- fatty acids like alpha-linolenic acid (αLA), eicosapentaenoic acid (EPA) and docosahexaenoic acid (DHA). The FFAR4 receptor can be found in the taste buds, liver, adipose tissue, intestines, macrophages, pancreas, osteoclasts and osteoblasts [50]. FFA4R receptor is a multifunctional protein that participates in insulin sensitivity, macrophage functions, hepatic steatosis and hormone secretion from the pancreatic islets or intestinal endocrine cells [51].

FFAR4 is overexpressed in colorectal cancer, and its activation enhanced cell migration and induced a pro-angiogenic response in colon cancer cells. Moreover, xenograft experiments in nude FFAR4-KO mice demonstrated that the pharmacological stimulation of FFAR4 promotes tumor growth [52]. Another study performed in a different murine model of colon cancer proposes the implication of FFAR4 on chemotherapy resistance by modifying the macrophage function [53] and, in the same line, high levels of FFAR4 in breast tumors were associated to a reduced response to chemotherapy in patients. In this case, in vitro and in vivo mechanistic studies, indicated that this receptor promoted cancer cell resistance to cytotoxic agents by increasing the expression of multidrug resistance proteins [54]. Finally, an oncogenic role for FFAR4 has also been observed in breast cancer, where it promotes migration and epithelial to mesenchymal transition in cells and metastasis in a xenograft model [55], and in pancreatic cancer cells, where FFAR4 stimulates cellular motility [49]. In contrast, experiments combining pharmacological stimulation and gene silencing, demonstrate an inhibitory effect of FFAR4 on migration of melanoma and prostate cancer cells, and on proliferation of the latter [46,56], while indirect evidences suggest similar inhibitory effects on lung and breast cancer cells [47,57]. Thus, data on patients and in vivo experiments indicate that FFAR4 may exert protumoral activities while some studies on isolated cells defend the opposite idea. New investigations are needed to determine if both concepts apply in the clinical setting and, in such case, whether the final role of FFAR4 depends on the kind of tumor.

#### 3.2.3. G-Protein-Coupled Receptor 84 (GPR84)

MCFAs, with a chain length of 9–14 carbons, bind GPR84 receptor [58]. It is predominantly expressed in immune system-related tissues and cells, such as bone marrow, spleen, lung, lymph nodes and brain microglia, and adipose tissue [59]. It is suggested that GPR84 is a proinflammatory receptor [60] and it could play a role in linking fatty acid metabolism and immune responses [61]. Moreover, it has been described that participates in the Wnt signaling in stem cells, promoting β-catenin signaling and leukemic stem cells maintenance [62]. GPR84 agonists might be useful for the treatment of cancer by activating the immune response (immuno-oncology), whereas desensitization of GPR84 by agonists might lead to a functional blockade of the receptor, resulting in anti-inflammatory effects and possibly antiproliferative effects on acute myeloid leukemia [62]. Finally, GPR84 has been linked to liver fibrosis that is a major risk factor in hepatocarcinoma and an independent risk factor of recurrence after hepatectomy [63,64].

#### 3.2.4. G-Protein-Coupled Receptor 119 (GPR119)

GPR119 is expressed predominantly in the pancreas (β-cells) and gastrointestinal tract (enteroendocrine cells) in humans. GPR119 can be activated by oleoylethanolamide and several other endogenous lipids containing oleic acid: these include N-oleoyl-dopamine, 1-oleoyl-lysophosphatidylcholine (generated in the tissue) and 2-oleoyl glycerol (generated in the gut lumen). GPR119 agonists may have a promising role in the treatment of type 2 diabetes and related metabolic disorders [65]. Although its relevance in cancer is practically unexplored, the putative role of GPR119 as a cannabinoid receptor allows to consider its modulation for symptom management and cancer therapy [66].

## 4. Hydroxycarboxylic Acid Receptors

This family of receptors includes GPR81 (HCAR1), GPR109A (HCAR2), and GPR109B (HCAR3), which share significant sequence homology and respond to endogenous hydroxy-carboxylic acid metabolites. The HCA_1_ receptor is activated by 2-hydroxy-propanoic acid (lactate), the HCA_2_ receptor is a receptor for the ketone body 3-hydroxy-butyric acid, and the HCA_3_ receptor is stimulated by the β-oxidation intermediate 3-hydroxy-octanoic acid [67]. In Table 4 we gather the both in vitro and in vivo functional studies describing the relevance of HCA_1_, HCA_2_ and HCA_3_ receptors and we have also specified which cancer was analyzed.

### 4.1. GPR81/HCA_1_ Receptor

HCAR1 was identified as the lactate receptor only a few years ago [68]. It is coupled to the Gi-protein and seems to present a restricted pattern of expression, with significant levels in white and brown adipocytes and relatively low expression in several other tissues like the liver, kidney, skeletal muscle, or gastric tissue [67].

This receptor may gain importance in tumors because cancer cells often present an abnormally high rate of aerobic glycolysis, and transform the glycolysis product (L-pyruvate) to lactate via lactate dehydrogenase (LDHA). Lactic acid is effluxed out of the cancer cells via the H^+^-coupled monocarboxylate transporter 1 (MCT1) and MCT4 to prevent intracellular acidification and, consequently, lactate accumulates in the extracellular space. Lactate stimulates the expression of its own receptor [69,70] and, maybe, because of that, high expression of HCAR1 has been detected in pancreatic tumors [71], in breast cancer [72,73] and in cell lines from colon, breast, lung, hepatocellular, salivary gland, cervical, and pancreatic carcinomas [71,72,73,74].

The presence of this receptor in cancer cells seems to promote their growth and survival. Silencing of HCAR1 in breast cancer cells reduces their proliferation rate and promote their apoptosis [72,73]. A similar effect was observed in pancreatic cancer cells although only when cultured with lactate as the unique energy source [71]. Additionally, HCAR1 seems to modulate the sensitivity of cancer cells to cytotoxic agents. In human cervical cancer HeLa cells, activation of HCAR1 increased DNA repair rate and promote cellular resistance to clinically used antineoplastic drugs [74,75,76]. Silencing HCAR1 in cancer cells also inhibits the growth of breast and pancreatic tumors in vivo [71,72], and in the last case, a reduced metastatic potential has been observed [72]. These actions may be related with direct and indirect HCAR1-mediated effects in tumor stromal cells. In this sense, the activation of this receptor in breast cancer cells promotes the release of pro-angiogenic factors while its silencing reduces angiogenesis in a xenograft mouse model, which was associated to a reduced rate of tumor growth [72]. Similarly, HCAR1 activation in lung cancer cells activated mechanisms that support the tumor immune evasion like an increased expression of the programmed death ligand 1 (PD-L1), a reduced production of interferon-γ or an increased apoptosis in cocultured Jurkat T-cell leukemia cells [77]. Along the same line, the activation of this receptor seems to promote a pro-tumor tolerogenic profile in tumor-recruited plasmacytoid dendritic cells [78] and its expression in dendritic cells derived from murine mammary gland tumors suppresses T-cell function.

In summary, all the reported evidences support a pro-cancerous role of the lactate-HCAR1 pathway and point to the blockade of this receptor as a potential anti-neoplastic strategy but unfortunately, no antagonists or blocking antibodies are available to be tested in this setting at present [79].

### 4.2. GPR109A/HCA_2_ Receptor and GPR109B/HCA_3_ Receptor

HCAR2 and HCAR3 are activated by intermediates of central metabolic processes that are often differentially regulated in cancer cells. Both receptors share a high structural homology, but their expression and relevance in cancer slightly differs.

HCAR2 was discovered in highly differentiated adipocytes, spleen, and immune cells [80]. In addition, this receptor is also detected in intestinal /colonic epithelial cells, in which the expression is restricted to the lumen-facing apical membrane [81]. The main ligands for HCAR2 are niacin and the ketone body, β-D-hydroxybutyrate, a hydroxycarboxylic acid metabolite generated from acetate and butyrate in epithelial cells. Upon binding to these ligands, HCAR2 couples to G proteins of the Gi family, and signals the decrease of cAMP.

It was early observed that the expression of HCAR2 was diminished in several human cancers such as colon, compared with the paired colon tissue [81,82] and primary breast tumor tissues [83]; in addition, an impaired functionality of HCAR2 was associated with skin cancer [80]. Recently, a reduced GRPR109A expression has also been detected in different murine models of colon cancer [84,85]. Several mechanisms have been involved in the cancer associated silencing of GPR109. DNA methylation epigenetic mechanisms, due to the increased expression levels of DNA methylases (DNMT) found in colon cancer cell lines and in primary colon cancer [81]. In addition, the posttranslational modification of this gene has also been suggested [17].

The functional relevance of HCAR2 has been stated in in vitro studies which show an anti-proliferative activity of this receptor. An increased proliferation was detected in breast cancer cells after the knock down of HCAR2 [73] while, the activation of HCAR2 by butyrate, induced the apoptosis of both colon cancer cells [81] and breast cancer cells [83]. Furthermore, in vivo studies have also demonstrated that the activation of HCAR2, with niacin and the commensal metabolite butyrate, suppresses colonic inflammation and carcinogenesis [86,87], while deletion of HCAR2 increased tumor incidence of spontaneous breast cancer in transgenic mice [83]. In contrast to data reported in colon and breast cancer, a recent study shows an increased proliferation of cells from glioblastoma by the activation of HCAR2 with butyrate [88]. A better knowledge of the specific tissue-conditioning factors of cancer is warranted to understand these discrepancies.

As a whole, strong evidence support that HCAR2 in the colon, acts as a tumor suppressor. It is important to note that, in a similar manner to that reported with FFAR2, the reduced expression of HCAR2 associated to colon cancer, difficult the potential beneficial effects of the activation of this receptor on tumor progression. Of interest, several strategies have been proposed as alternative therapeutic pathways to increase the expression of HCAR2 in colon cancer; the concomitant administration of SCFAs with DNMT inhibitors, such as procainamide, [81] as well as the administration of natural products such as

In contrast to HCAR2, HCAR3 is only expressed in humans and higher primates [89] and it has been mainly detected in adipocytes. HCAR3 is activated by the b-oxidation intermediate, 3-hydroxy-octanoic acid and it signals through Gi proteins. In a similar manner to that reported for HCAR2, the functionality of HCAR3 seems to be impaired in skin cancer [80], but little is known about its relevance in tumor processes.

## 5. GPCRS for Amino Acids and Related Metabolites

Amino acids (AAs) are organic molecules composed of an acidic carboxyl group (-COOH), a basic amino group (-NH2) and an organic R group which is unique to each amino acid. Although more than 300 amino acids have been described in nature, only 20 can be structural units of proteins. In the oncologic environment, the metabolism of amino acids guarantees the required energy for the cellular proliferation, differentiation and redox equilibrium [90]. Given the catabolism of AAs, a wide spectrum of several amino acid-derived metabolites has been described. Out all of GPCRs, four receptors can be selectively activated by amino acid-derived metabolites: Calcium-sensing receptor (CasR), Trace amine associated receptor 1 (TAAR1), GPR35 and GPR142. In this section of the review, we will describe the role of these GPCRs in different types of cancer (Table 5) and we will discuss whether they can be considered as pharmacological targets for cancer treatment.

### 5.1. Calcium-Sensing Receptor (CasR)

The extracellular calcium-sensing receptor (CasR) is a G-protein-coupled receptor belonging to the class C of the GPCR family which is mostly expressed in calcitropic tissues such as kidney and parathyroid glands. Additionally, it can also be found in lungs, skin, intestine, brain, and vasculature [91].

CasR can be activated by a myriad of different types of ligands that include L-amino acids, cations, polyamines, glutamyl peptides and some anions such as PO_4_^3−^ and SO_4_^2−^. The activation of this receptor can stimulate several signaling cascades mediated by G_q/11_, G_i/o_, and G_12/13_ proteins [92] and, besides its implication in the Ca^2+^ homeostasis, it has been associated with a number of different functions that include cell proliferation, differentiation and apoptosis, as well as regulation of enteroendocrine hormone secretion, vascular tone, lung and neuronal development, or cardiac function [91].

In cancer, both, increased and decreased expression has been reported. While CasR is overexpressed in prostate and metastatic breast cancer, it is reduced in some neuroblastic tumors and colorectal cancer [93]. An oncogenic role of CasR has been described in breast, renal, prostate and gastric cancers. Regarding breast cancer, some evidences support an association between three different Single Nucleotide Polymorphisms (SNPs) in CasR (rs112594756, rs17251221, and rs1801725) and breast cancer risk, higher aggressiveness and unfavorable outcomes [94,95]. In fact, CasR stimulates the proliferation of breast cancer cells and mediates the promoting effect of extracellular calcium on bone metastasis [96]. In line with this, the overexpression of this receptor enhances the osteolytic potential of intratibially injected breast cancer cells through epiregulin-mediated osteoprotegerin downregulation [97]. The implication of CasR in bone metastasis was also observed in renal cell carcinoma. Frees and colleagues demonstrated that calcium-stimulated CasR increased cell migration and proliferation in human 768-O renal cell carcinoma cells and, in a xenograft mouse model in vivo, the injection of cells overexpressing CasR increased bone metastasis [98].

CasR is significantly expressed in advanced and aggressive prostate cancers, such as metastatic castration resistant tumors and neuroendocrine prostate cancers, and a correlation between high CasR expression and decreased survival was observed [99]. The possible therapeutic effect of blocking this receptor has been recently reported by Yamamura and colleagues, who demonstrated that the proliferation and migration of human prostate cancer cells was impaired by the CasR antagonist calcilytics [100].

A protumoral effect of CasR has also been reported in gastric cancer. CaSR expression was enhanced in gastric cancer specimens and positively correlated with serum calcium concentrations, tumor progression and poor survival. The same study showed that CasR mediates the Ca^2+^/AKT/β-catenin pathway that increases proliferation, migration, and invasion in isolated gastric cancer cells and, in a xenograft model, this receptor was involved on the aggravating effect of the local injection of calcium in gastric tumor growth and metastasis [101]. In addition, the same group has stablished a functional linkage between CasR and the oncogene telomerase reverse transcriptase in the development of gastric cancers [102]. The pharmacological potential of CasR is supported by a recent publication reporting that the NPS-2143, a CasR antagonist, reduces the proliferation, migration and invasion and promotes the apoptosis of the AGS cells [103].

In contrast to all of this, and in part due to the growing evidence pointing to a beneficial effect of calcium against colon carcinoma, CasR has been also proposed as a tumor suppressor in this type of cancer. In fact, Momen-Heravi and colleagues reported a reduced expression of CasR in colorectal patients and an association between higher expression of CasR and lower risk of mortality, suggesting that this receptor might be a biomarker for good prognosis [104]. This lower expression seems a consequence of epigenetic alterations since a huge number of CpG islands of the CasR gene are highly methylated in colorectal tumors. Moreover, the presence of some miRNAs, such as miR-21, miR-135a, miR-135b, miR-145, miR-146b, and miR-503, reduced the expression of CasR in colorectal tumors [105]. The relevance of CasR in colorectal tumorigenesis was demonstrated in the conditional knock-out of CasR in intestinal epithelium which presented intestinal hyperproliferation, developed pre-malignant lesions and were more susceptible to azoxymethane. These effects were associated to an increased translocation of β-catenin into the nucleus and the activation of the proliferative Wnt signaling pathway [106]. In addition, it was also reported that this receptor can also activate the non-canonical Wnt pathway through Wnt5a and its receptor Ror2 to promote colonic differentiation and inhibit proliferation [107].

An anti-tumoral effect of CasR has also been observed in neuroblastoma. In this case, both a lower CaSR expression due to gene hypermethylation and the presence of two polymorphisms, rs7652589 and rs1501899 have been associated with neuroblastic tumors [108]. Interestingly, the expression of CasR correlated with positive prognostic variables such as low clinical stage, low clinical risk, age at diagnosis and differentiated histology [109]. In line with these clinical evidences, the overexpression of CasR in neuroblastoma cell lines significantly reduced the proliferation and activated apoptosis through ERK1/2, while the calcimimetic cinacalcet reduced the growth of neuroblastoma due to a promotion of the differentiation, ER stress and apoptosis [110].

Altogether, it is clear that CasR plays a yin–yang role in cancer, which makes essential to further elucidate the role of this receptor in each cancerous process. This dual role makes more difficult to propose a pharmacological strategy against this receptor in order to treat several type of cancers. Obviously, further studies are needed to clarify whether CasR can be a valid pharmacological target.

### 5.2. Trace Amine Associated Receptor 1 (TAAR1)

Trace amine associated receptor 1 (TAAR1) is expressed in pancreas, intestine, stomach and the central nervous system and is involved in the regulation of the classical monoamine neurotransmission [111]. TAAR1 can be activated by different endogenous molecules called trace amines (TAs) that can be found in the mammalian nervous system and whose roles may overlap with those of the classical monoamines. Whereas TAAR1 presents low affinity for classic monoamines, it can be activated with nanomolar or micromolar levels of some endogenous amines such as β-phenylethylamine, p-tyramine, octopamine and tryptamine, which are synthetized after the decarboxylation of aromatic amino acids [112]. In addition, TAAR1 can be stimulated by methamphetamines and ergolines. Due to the nature of the TAAR1 agonists and the imbalance of TAs detected in several pathological scenarios, this receptor has been suggested as a novel target for central pathologies such as depression, schizophrenia, and drug addiction [113].

The binding of a specific agonist with TAAR1 leads to the activation of several signal cascades which have not been fully well-characterized. In fact, this receptor could couple with Gs protein, triggering the activation of adenylate cyclase, although other G proteins, including Gq and Gα16, could also be activated by this receptor [114]. On the other hand, TAAR1 signaling also includes a G-protein-independent / b-arrestin2–dependent pathway [115].

Recently, the role of TAAR1 in different cancers has become a topic of emerging interest. Fleischer et al. analyzed different public databases of RNA-sequencing information and detected a differential TAAR1 gene expression in breast, bladder, cervical, lung, pancreatic, stomach, renal, and thyroid cancer. In addition, they also reported both a TAAR1-downregulation of in sarcoma, renal, cervical, liver, kidney, pancreas, prostate, pituitary, and uterine cancers, and an upregulated expression of TAAR1 in esophageal, stomach and lung cancers [116].

Although the differential pattern of TAAR1 expression in diverse cancers would suggest that the role of this receptor might be different depending on the cancer type, the limited number of functional studies performed so far point to the activation of TAAR1 as a possible anti-neoplastic strategy. In leukemia, lymphoma and other B-cell pathologies, TAAR1 has been proposed as a possible pharmacological target since its activation with several agonists induced apoptosis in L3055 Burkitt lymphoma cells and this cytotoxicity seemed to be specific for malignant B cells since normal B cells were less sensitive to those agonists [117]. On the other hand, in breast cancer, Vattai and colleagues reported that TAAR1 expression correlates with the tumor differentiation grade, so an overexpression of TAAR1 was associated with a significantly longer survival [118]. In line with this, Kovacks and colleagues have recently reproduced the same observations in patients with both estrogen receptor (ER) negative and ER + breast cancer and observed that the TAAR1 agonist cadaverine exerts a beneficial effect against the development of breast cancer in vivo through TAARs [119]. Therefore, the available evidences at present suggest that TAAR1 might represent a relevant pharmacological target against some cancers, which makes essential the development of specific TAAR1 agonists.

### 5.3. G-Protein-Coupled Receptor 35 (GPR35)

The G-protein-coupled receptor 35 (GPR35) is an orphan G protein-coupled receptor (GPCR) which is highly expressed in the digestive tract, skeletal muscle, lung, uterus, and dorsal root ganglion, while a moderate expression of GPR35 has also been detected in liver, heart, spinal cord, bladder, brain and cerebrum. Additionally, this receptor is expressed in different immune cells such as peripheral monocytes, basophils, eosinophils, mast cells, and invariant natural killer T (iNKT) cells [120].

GPR35 can be activated by different compounds. The first reported agonist was the kynurenic acid (KYNA) which can activate this receptor at high levels. In addition, it has been recently described that GPR35 has a high affinity for the chemokine CXCL17, although additional studies are needed to confirm whether this mucosal chemokine is an endogenous ligand of this receptor. Apart from these putative endogenous ligands, some synthetic compounds such as zaprinast, pamoic acid, furosemide and cromolyn have also been identified as GPR35 agonists [121]. KYNA is an endogenous metabolite produced in one branch of the kynurenine pathway of tryptophan metabolism. This neuroprotective metabolite is produced by astrocytes and inhibits the three classes of ionotropic excitatory amino acid receptors. The binding of KYNA with GPR35 receptor triggers several molecular mechanisms such as: a reduction of cAMP and calcium levels, the inhibition of the phosphorylation of AKT, ERK, and p38 and an increase in the β-catenin levels. Interestingly, all of these molecular effects lead to a reduction of inflammation, since increased calcium levels are associated with the activation of the inflammatory pathway and the activation of NF-κB [120].

Several studies report alterations in the concentration of KYNA and the expression of GPR35 in different cancers. Increased levels of KYNA have been detected in colon carcinoma, oral squamous cell carcinoma, non-small cell lung carcinoma and multiple myeloma, whereas a reduced concentration of this metabolite was observed in renal cell carcinoma and primary cervical cancer [122]. In parallel, and in comparison to paired normal tissues, the expression of GPR35 was reduced in prostate, testicular and thyroid tumors; increased in stomach, pancreatic, colon and non-small-cell lung cancer, and unaltered in breast and ovarian cancer [122]. Functionally, GPR35 signaling pathway via ERK kinase has been involved in several cellular processes such as proliferation, cell survival and even metastasis [123], and overexpression of GPR35 seems to confer drug resistance in non-small-cell lung cancer through β-arrestin-2/Akt signaling [124]. More recently, an elegant study by Schneditz and colleagues showed that the loss of GPR35 or its inhibition by a selective pepducin prevented the inflammation-associated and spontaneous intestinal tumorigenesis [125]. Thus, the existing functional studies present GPR35 as an oncogenic receptor. However, the fact that in some cancers GPR35 or its ligand are increased whereas both of them are decreased in others lead us to suggest that this receptor might play a dual role in carcinogenesis, an issue that should be investigated to elucidate the specific role of this receptor in particular cancers. Besides its potential as a pharmacological target, a recent study suggests that the expression of a GPR35 splice variant (GPR35 V2/3) in colon carcinoma patients might be useful as an indicator of poor prognosis [126], although further studies should be performed to establish its value as a prognostic factor.

### 5.4. G-Protein-Coupled Receptor 142 (GPR142)

G-protein-coupled receptor 142 (GPR142) is a Gq-coupled GPCR, which is highly and almost exclusively expressed in pancreatic β, α, and enteroendocrine cells [127]. Although with a lower expression, GPR142 has also been detected in stomach, the duodenum, the ileum, and the jejunum [128]. GPR142 is a highly selective sensor of essential aromatic amino acids, specifically L-Tryptophan (L-Trp) and L-Phenylalanine, being the L-Tryptophan the most efficacious and potent agonist. The binding of L-Trp triggers the activation of both Gq and Gi-coupled signaling and the activation of ERK. Since L-Trp stimulates insulin secretion and improves glucose tolerance, GPR142 was rapidly associated to a potential role in the regulation of glucose homeostasis and in metabolic diseases such as obesity or diabetes [129]. This has motivated the design of synthetic GPR142 agonists, one of this has recently reached phase 1 in clinical trials for Type 2 diabetes treatment [130].

In spite of the huge research performed on the role of this receptor in Type 2 diabetes, the relevance of this receptor in cancer has not been studied yet. As far as we know, there is only one study linking one missense mutation in GPR142 gene with type 1 and type 2 papillary renal cell carcinoma, and suggesting therefore that GPR142 might be involved in this type of cancer [131]. Aside from this, we should consider the possibility of GPR142 being an indirect mediator in the numerous kinds of cancer, such as pancreatic, endometrial, breast, liver, colorectal, bladder and kidney cancer, that have been positively associated with type 2 diabetes [132].

## 6. Bile Acid-Sensing GPCRs

Bile Acids (BAs) are hydroxylated steroids synthesized in the liver initially as cholic acid and chenodeoxycholic acid, and then conjugated with taurine and glycine in order to be secreted into the bile. Besides their role favoring the intestinal absorption of lipids, these steroids also exert metabolic effects by signaling through several nuclear receptors (farnesoid X receptor, vitamin D receptor, pregnane X receptor), and through the G-protein-coupled bile acid receptor-1 (GPBAR-1) [133].

Alterations in the synthesis, secretion and absorption of BAs exert harmful local and systemic effects triggering the activation of inflammation, metabolic disorders, liver pathologies and even cancer. In fact, BAs can act as cancer-promoting agents due to the regulation of the proliferation of cancer cells from multiple origin. In recent years, accumulative evidence has described the pathways involved in the carcinogenic properties of BAs including DNA damage and genomic stability, oxidative stress, epigenetic factors, apoptosis and activation of nuclear receptors [134]. In this section of this review, we will describe the relevance of the GPCR which can be activated specifically by BAs, GPBAR-1, in different types of cancer (Table 5) and we will discuss its potential therapeutic role.

### G-Protein-Coupled Bile Acid Receptor-1 (GPBAR-1)

G-protein-coupled bile acid receptor-1 (GPBAR-1), also known as Takeda G protein-coupled receptor 5 (TGR5), can be selectively activated by both unconjugated and conjugated primary or secondary BAs. TGR5 is expressed ubiquitously distributed throughout the body in human tissues. Indeed, high levels of TGR5 mRNA have been detected in the gallbladder, spleen, liver, placenta, lung, liver, intestine, adipose tissue, smooth muscle, kidney, female reproductive organs and fetal kidney and liver. In most of the cells, this receptor couples to a stimulatory G alpha protein (Gα_s_), so that after its activation it activates the adenylate cyclase and increases the intracellular cyclic AMP. Nevertheless, in cholangiocytes TGR5 can couple to either Gα_s_ or an inhibitory G alpha protein (Gα_i_) depending on its subcellular localization [135].

There is accumulating evidence about the role of TGR5 in different cancers since an altered expression or activity of this receptor affect several signaling pathways implicated in cancer formation [136]. In human gastric carcinoma cells (AGS), the activation of TGR5 by deoxycholic acid triggers the activation of ERK1/2, MAPK, and the epidermal growth factor receptor (EGF-R), and its silencing promotes apoptosis [137]. In fact, the expression of this receptor in gastric cancer is increased and positively correlates with the epithelial-to-mesenchymal transition (EMT), a crucial process involved in metastasis [138]. Moreover, an enhanced expression of TGR5 is associated with a lower life expectancy in patients with esophageal and gastric adenocarcinomas [139]. The activation of this receptor has also been associated to cholangiocarcinoma progression by increasing proliferation, migration, and mitochondrial metabolism [140]. Finally, TGR5 is aberrantly overexpressed in non-small cell lung cancer, where its activation promotes cell proliferation and migration through JAK2 and STAT3 pathway [141]. In the field of diagnosis, Zhao and colleagues propose the use of the specific reduction in GPBAR-1 observed in clear cell renal carcinomas, as a distinguishing factor from other malignant renal carcinomas such as papillary renal carcinoma cells (RCCs), chromophobe RCCs, collecting duct carcinomas or clear cell papillary RCCs [142].

In conclusion, the protumoral role of TGR5 observed in different types of cancer highlights this receptor as a promising pharmacological target. However, no clinical trial is at present analyzing the anti-neoplastic potential of the available TGR5 antagonists.

## 7. pH-Sensitive Receptors

The family of proton-sensing GPCRs include four members: GPR4, GPR65, or T-cell death-associated gene 8 (TDAG8), GPR68, or Ovarian cancer G protein-coupled receptor 1 (OGR1) and GPR132 or G2A. They have been identified as pH sensors that respond to extracellular acidosis through the protonation of several histidine residues. GPR4 signaling involves the G12/13/Rho, the Gq/PLC, and the Gs-protein/cAMP pathways; GPR65 is linked to Gs/cAMP pathway; GPR68 is coupled with the PLC/Ca^2+^ signaling through Gq/11 proteins while GPR132 signals through Gi/o/cAMP and Gq/11/ Ca^2+^ [143,144,145].

These receptors may be of significance in cancer because tumors often present an acidic extracellular environment due to the abnormal metabolic activity of cancer cells. The reduced pH is mainly the consequence of the secretion of lactate and H^+^, and the conversion of the CO_2_ generated in the tricarboxylic acid cycle into H^+^ and HCO_3_ in the extracellular space [146,147]. In the following section of this review, we will describe the role of the four members of proton-sensing-GPCRs in different types of cancer. In Table 6 we have included all the in vitro and in vivo functional studies which demonstrate a beneficial or harmful effect of each proton-sensing GPCR in any kind of human cancer.

### 7.1. GPR4

Initial northern blot analysis detected a high physiological GPR4 expression in lung, and lower levels in heart, kidney skeletal muscle, liver, and pancreas [148]. In cancer, GPR4 gene overexpression was initially detected in a variable proportion of breast, ovarian, colon, liver and kidney tumors, but not in lung or prostate tumors [149]. More recently, the analysis of public RNA sequencing data on 23 different kinds of tumors showed that most cancers present a differential expression of GPR4, but only a few reached statistical significance. GPR4 appeared up-regulated in cholangiocarcinoma, down-regulated in cervical and lung cancers, and increased or decreased in kidney tumors depending on the kind of cancer [150]. Analysis centered in particular cancers revealed the up-regulated expression of GPR4 in head and neck squamous cell carcinoma [151], in renal cell carcinoma [152], in colorectal cancer [153] and in hepatocellular carcinoma [154] and, in most of these studies, its high expression correlated with late stage tumors and poor overall survival [152,153,154,155].

Functional studies show divergent roles of GPR4 in different types of cancer. On one hand, GPR4-knock out (KO) mice showed reduced tumorigenesis after the orthotropic transplantation of cells from murine breast (4T1) or colon cancer (CT26) cell lines [156] while suppression of GPR4 in human colorectal cancer (HCT116) cells significantly attenuated tumor growth of subcutaneous xenografts and reduced liver invasion in a metastasis model in nude mice [153]. This protumoral effect may be related with a proangiogenic activity since colonic tumors from GPR4-KO mice presented an altered vessel morphology, length and density [156] and GPR4-transfected cells promoted the formation of tubes by human microvascular endothelial cells in a paracrine manner in vitro [151]. On the other, GPR4 overexpression in murine melanoma cells reduced their migratory ability in vitro and suppressed pulmonary metastasis when injected intravenously in mice, although it did not affect the formation of a primary tumor when these cells were administered subcutaneously [157,158]. Recent efforts aimed at developing GPR4 modulators [159,160] have given rise to an specific orally-active compound that will probably help to characterize the potential of this receptor as a new therapeutic target against cancer.

### 7.2. G-protein-Coupled Receptor 65 (GPR65)

GPR65 expression has been detected in immune cells, including T and B lymphocytes, neutrophils, eosinophils, and mast cells, and in spleen, lymph nodes, thymus, lung, and small intestine [145]. With regard to its presence in tumor tissues, its overexpression has been detected in a proportion of kidney, ovarian, colon and breast tumors [149] and in metastatic melanomas [161] while, in contrast, a reduced expression of GPR65 has been observed in multiple hematological malignancies [162,163].

The role of GPR65 in the cancerous process is not well characterized and the present evidences describe a complex picture. A protumoral role of this receptor can be deduced from experiments demonstrating that its overexpression in normal mammary epithelial cells (NMuMG), in mouse Lewis (LLC) or human (A549) lung carcinoma cells significantly augments their carcinogenic potential in vivo [149,164]. In parallel, GPR65 silencing in human non-small cell lung cancer cells (NCI-H460), which constitutively express high levels of this receptor, significantly reduced tumor development in nude mice [164]. GPR65 promotes their survival and growth in acidic conditions [164], which would mean an advantage in the tumor environment. In parallel, in T cell lymphoma cell lines, GPR65 mediates the protective effect induced by acidosis against multiple metabolic stresses by increasing the expression of the anti-apoptotic proteins Bcl2 and Bcl-xL [165,166]. However, other experiments performed in the same murine lymphoma cell line (WEHI7.2) show that GPR65 overexpression induces apoptosis while its silencing prevents the pro-apoptotic effect of glucocorticoids [167]. Similarly, transfection of GPR65 in human lymphoma cells (U937) decreases the expression of the oncogene c-Myc and their proliferative activity, and contrary effects occur when GPR65 is knockdown [162,163], and in vivo, these GPR65-transfected U937 cells showed a reduced tumorigenic and metastatic potential when xenografted in severe combined immunodeficient mice (SCID) mice [163].

Thus, different data suggest that the presence of GPR65 in tumor cells has the potential to affect the carcinogenic process, which seems logical as this receptor reacts to the extracellular acidosis commonly encountered in tumors. However, the divergent directions of the accumulated evidences suggest that the role of GPR65 in cancer may greatly depend on each specific context. Although some pharmacological modulators of GPR65 have been described [168,169], their role in the cancer context has not been analyzed.

### 7.3. G-Protein-Coupled Receptor 68 (GPR68)

GPR68 presents a widespread pattern of expression in normal human tissues [170] while the analysis of public RNA sequencing data of 45 different types of cancer reveals a significantly differential expression in more than 70% of these tumors. Increased expression was encountered in pancreatic, cervical, bladder, breast, ovarian, testicular, colon, and lung cancers, while a down-regulation was found in tumors affecting the prostate or the kidney. Tumors of the thyroid gland, the stomach or the esophagus showed increased or decreased expression of GPR68 depending on the type of cancer [171]. Other studies showed GPR68 overexpression in human medulloblastoma tissue [172] and in head and neck squamous cancer [173]. Finally, immunohistochemical analysis detected significant GPR68 expression in endocrine or neuroendocrine tumors with origin in the pituitary, the thyroid gland, or the respiratory and digestive systems [174]. More detailed analysis on the expression of GPR68 in particular tumor cell types, showed a significant expression in cancer epithelial cells and fibroblasts of pancreatic ductal adenocarcinomas [171] and in hematopoietic and mesenchymal cells of colorectal tumors [175].

Some studies suggest that the expression of GPR68 in cancer cells reduce their malignancy. In this sense, genetically induced overexpression of GPR68 in metastatic human prostate cancer cells (PC3) reduced the metastasis after their orthotopic transplantation in nude mice [176] while, in vitro, transfection of GPR68 to breast cancer cells reduced their migratory ability [177] and the induction of this receptor by lenalidomide in myelodysplastic syndrome cells mediated the pro-apoptotic effect of this antineoplastic agent [178]. In contrast to all this, the stromal expression of GPR68 seems to play a significant pro-tumoral effect. GPR68 deficiency in mice reduced the size of tumors induced by xenotransplantation of cells from murine melanoma (B16-F10) or prostate cancer (TRAMP-C2, RM-9) cell lines and, in both cases, an altered macrophagic response was observed [179,180]. In the case of prostate cancer, the administration of wild type macrophages normalized the tumor growth, thus suggesting that GPR68 expression in host cells contributes to tumor induced immunosuppression. A similarly reduced tumorigenic response was observed in GPR68 KO mice injected with colon cancer (MC-38) cells but without a significant alteration in the immune response [175]. However, the same study shows that the expression of GPR68 in fibroblasts promotes the formation of tumor spheroids by human colorectal carcinoma cells (HCT116) in vitro, and the same occurs with the in vivo growing of cells from human metastatic breast carcinoma (MDA-MB-453) in nude mice [181]. In this line, Wiley et al. described a loop between cancer cells and fibroblasts from pancreatic tumors by which cancer cells promote GPR68 expression in cancer-associated fibroblast, whose activation promotes cancer cell proliferation through interleukin-6 (IL6) release [171]. All these results suggest that GPR68 may play beneficial or deleterious effects on cancer progression depending on the kind of cells in which is expressed. Some evidences demonstrate that GPR68 activity can be modulated by different benzodiazepines acting as allosteric regulators able to bias the pattern of activation towards particular signaling pathways [182,183]. This constitutes a complex and incipient field of research in the search for new GPR68-targeting antineoplastic strategies.

### 7.4. G-Protein-Coupled Receptor 132 (GPR132)

GPR132, also known as G2 accumulation protein (G2A), is highly expressed in macrophages, hematopoietic tissues rich in lymphocytes, like spleen and thymus, as well as in lung and heart [145], and has been involved in the regulation of leukocyte functions and inflammation [184].

Initial studies analyzing the putative role of GPR132 in cancer obtained opposite results on its influence of fibroblast transformation [185,186]. However, this receptor seems to exert an antitumoral effect in blood cancers. It was initially observed that expression of GPR132 inhibited the ability of the oncogene BCR-ABL to induce cellular growth in B-cell precursors [185], and a more recent study demonstrates that GPR132 deficiency promotes leukemogenesis in mice receiving BCR-ABL transduced bone marrow cells [187]. Additionally, pharmacological stimulation of GPR132 has shown inhibitory effects in a murine model of acute myeloid leukemia [188]. In contrast, two different studies suggest that the expression of this receptor in macrophages promotes cancer. Cheng et al used genetic and pharmacological approaches to demonstrate, in an orthotopic transplantation model of breast cancer, that GPR132 exerts a pro-tumoral effect by modulating the macrophage function, and propose that the anticancerous effect of the PPARγ agonist rosiglitazone depends on the down-regulation of this receptor [189]. Further supporting this idea, Chen et al have observed in an analogous model of breast cancer that GPR132 favors a protumoral phenotype in tumor-associated macrophages and promotes lung metastasis [190]. In consequence, experimental studies strongly suggest that GPR132 is relevant in cancer and point to opposite effects in different neoplasias. However, there is a remarkable absence of analysis in human tumors.

## 8. GPCRs for Citric Acid Cycle Intermediates

### G-Protein-Coupled Receptor 91 / Succinate Receptor 1 (GPR91/SUCNR1)

SUCNR1 is stimulated by the Krebs cycle intermediate succinate, and acts as a sensor of the presence of this metabolite in the extracellular space. SUCNR1 interacts with multiple G-proteins that include Gi/o and Gq, and, initially detected in kidney, liver, spleen, and adipose tissue, later studies have demonstrated a widespread pattern of expression of SUCNR1 in human tissues and cell types [186,191]. It is widely assumed that succinate accumulates in the tumor environment as a consequence of mutations in the genes encoding the succinate dehydrogenase subunits [192], and that succinate induces oncogenic effects through epigenetic modifications and the activation of hypoxic signaling [193]. However, very little is known about the expression and role of SUCNR1 in cancer.

Unfortunately, there is still a scarce number of functional studies analyzing the specific role of SUCNR1 in different types of cancer. Indeed, in Table 6 we summarize the two cancers where a precancerous effect of this receptor has been reported. One recent study shows that the expression of SUCNR1 is increased in human lung tumors, and demonstrates that its presence in lung tumor cells (A459) favors lung metastasis in mice. The authors suggest that, besides a direct effect on cancer cells, activation of SUCNR1 in macrophages contributes to the metastatic process by inducing a tumor-associated macrophage phenotype that promotes cancer cell migration through a paracrine action [194]. In colon cancer cells, SUCNR1 activation by succinate triggered the activation of EMT and promote their motility and migratory capacity by stimulating the Wnt signaling pathway [195]. These recent evidences together with the facts that SUCNR1 is expressed in significant stromal components such endothelial cells and fibroblasts, where it mediates angiogenic and fibrogenic responses [196,197], and that SUCNR1 expression can be up-regulated by the accumulation of succinate in the tumor environment, make likely that this signaling pathway would significantly affect the cancerous process. However, this field is in clear need of further research aimed to define the value of this receptor as a therapeutic target in the neoplastic process.

## 9. Conclusions

G-protein-coupled receptors constitute a large family of cell surface receptors that detect a wide range of different molecules outside the cell and trigger numerous molecular pathways. Out of all GPCRs described so far, we have focused our attention on those GPCRs that can be activated by different metabolites. These newly recognized signaling mediators have been organized in 6 groups: fatty acids, hydroxycarboxylic acids, amino acids, bile acids, protons and citric acid cycle intermediates.

Interestingly, a differential expression of metabolite-sensing GPCRs has been detected in several human tumors compared with their respective healthy tissues. A myriad of situations is observed with some receptors overexpressed, other repressed and most of them, up-regulated or down-regulated depending on the type of cancer. In some cases, the deregulated expression of these GPCRs depends on the stage of the disease. This differential expression has led researchers to consider some of these receptors as possible biomarkers for cancer disease and have postulated its use as a diagnostic criterion. In addition, in some cancers, different SNPs and mutations in GPCR genes have also been associated with a better or a worse prognosis, which shows up the utility of studying their genetics. Nevertheless, further studies are needed to confirm these observations by reproducing them in bigger cohorts of patients, and to test whether their diagnostic value applies to different cancer types.

GPCRs have always been attractive candidates as possible pharmacological targets. Despite the fact that we are accumulating evidences pointing to specific roles of some GPCR in particular types of cancer, there is still much to do in this field. It is important to highlight that, as it happens with their expression, most metabolite-sensing GPCRs show pro-cancerous or anti-cancerous effects depending on the tumor studied. Thus, although numerous functional in vitro and in vivo studies using synthetic agonists, antagonists, or genetic manipulations have revealed promising results in particular neoplastic contexts, the duality attributed to most of these receptors is without doubt a matter of concern and complicates their definition as potential therapeutic targets. Probably because of this, and the recent history of most of these GPCR, none of these options has been translated to clinical trials thus far. Therefore, much further research work is needed to better characterize the role of each metabolite-sensing GPCR in each cancer and, subsequently, develop new specific antineoplastic therapies targeted to these receptors.

## Figures and Tables

**Table 1 cells-09-02345-t001:** Classification of metabolite sensing-G-protein-coupled receptors (GPCRs) according the nature of their specific agonists. This table includes the specific endogenous agonists reported so far, the G proteins linked to each receptor, and the information about the corresponding gene (approved symbol, aliases, and name of the gene group) [16]. (FFA, free fatty acid; FFAR, free fatty acid receptor; HCA, hydroxycarboxylic acid; TA, trace amines; CaS receptor, calcium sensing receptor)

IUPHAR NAME	Respond to:	ENDOGENOUS AGONISTS	TRANSDUCER	EFFECTOR RESPONSE	INNITIAL SYMBOL	*APPROVED SYMBOL*	*ALIASES*	GENE GROUPS
FFA1 receptor	Fatty acids	Docosahexaenoic acid, A-linolenic acid, Oleic acid, Myristic acid	Gq/G11 family	Adenylate cyclase inhibition Phospholipase C stimulation	GPR40	*FFAR1*		Free fatty acid receptors
FFA2 receptor	Fatty acids	Propanoic acid, Acetic acid, Butyric acid, Trans-2-methylcrotonic acid, 1-methylcyclopropanecarboxylic acid	Gq/G11 family Gi/Go family	Phospholipase C stimulation Adenylate cyclase inhibition	GPR43	*FFAR2*	*FFA2*	Free fatty acid receptors
FFA3 receptor	Fatty acids	Propanoic acid, Butyric acid, 1-methylcyclopropanecarboxylic acid	Gi/Go family	Adenylate cyclase inhibition	GPR41	*FFAR3*	*LSSIG*	Free fatty acid receptors
FFA4 receptor	Fatty acids	A-linolenic acid, Myristic acid, A-linolenic acid	Gq/G11 family β-arrestin2		GPR120	*FFAR4*	*PGR4*	Free fatty acid receptors
GPR84	Fatty acids	decanoic acid, undecanoic acid, lauric acid	Gi/Go family		GPR84	*GPR84*	*EX33*	GPCR-Class A orphans
GPR119	Fatty acid related compounds	N-oleoylethanolamide, N-palmitoylethanolamine	Gs family	Adenylate cyclase stimulation	GPR119	*GPR119*	*GPCR2*	GPCR-Class A orphans
HCA_1_ receptor	Hydroxycarboxylic acids	L-lactic acid	Gi/Go family		GPR81	*HCAR1*	*GPR104, HCA1*	Hydroxy-carboxylic acid receptors
HCA_2_ receptor	Hydroxycarboxylic acids, fatty acids	β-D-hydroxybutyric acid, butyric acid	Gi/Go family	Adenylate cyclase inhibition Phospholipase A2 stimulation	GPR109A	*HCAR2*	*HCA2, HM74A, NIACR1*	Hydroxy-carboxylic acid receptors
HCA_3_ receptor	Hydroxycarboxylic acids	3-hydroxyoctanoic acid	Gi/Go family	Adenylate cyclase inhibition	GPR109B	*HCAR3*	*HCA3, HM74B, NIACR2*	Hydroxy-carboxylic acid receptors
CaS receptor (provisional)	Amino acids, cations, small peptides, polyamines	L-phenylalanine, L-tryptophan, L-histidine, L-alanine, L-serine, L-proline, L-glutamic acid, L-aspartic acid, Gd^3+^, Ca^2+^, Mg^2+^, S-methylglutathione, γGlu-Val-Gly, glutathione, γGlu-Cys, spermine, spermidine, putrescine, PO_4_^3−^ and SO_4_^2−^	Gi/Go family Gq/G11 family G12/G13 family	Adenylate cyclase inhibition Phospholipase C stimulation Phospholipase D stimulation Adenylate cyclase stimulation Potassium channel Phospholipase A2 stimulation Phospholipase D stimulation	GPRC2A	*CASR*	*PCAR1*	Calcium sensing receptors
TA_1_ receptor	Trace amines	Tyramine, β-phenylethylamine, octopamine, dopamine, 3-iodothyronamine	Gs family Gq and Gα16 β-arrestin2	Adenylate cyclase stimulation	-	*TAAR1*	*TA1, TAR1, TRAR1*	Trace amine receptors
GPR35	L-tryptophan derived metabolites and others	kynurenic acid, 2-oleoyl-LPA, cGMP, DHICA, Reverse T3, CXCL17	G(qi/o) family β-arrestin2	Calcium mobilization and inositol phosphate production	GPR35	*GPR35*	*KYNA Receptor*	GPCR-Class A orphans
GPR142	Aromatic amino acids	L-tryptophan	Gq family		GPR142	GPR142	*PGR2*	GPCR-Class A orphans
GPBA receptor	Bile acids	lithocholic acid, deoxycholic acid, chenodeoxycholic acid, cholic acid	Gs family	Adenylate cyclase stimulation	GPR131	*GPBAR1*	*BG37, TGR5, M-BAR, GPCR19, MGC40597*	G protein-coupled bile acid receptor
GPR65	Protons	Protons	Gs family	Adenylate cyclase stimulation	GPR65	*GPR65*	*TDAG8*	GPCR-Class A orphans
GPR68	Protons	Protons	Gi/Go family Gq/G11 family	Adenylate cyclase inhibition Phospholipase C stimulation	GPR68	*GPR68*	*OGR1, GPR12A*	GPCR-Class A orphans
GPR4	Protons	Protons	Gs family Gi/Go family Gq/G11 family G12/G13 family	Adenylate cyclase stimulation Phospholipase C stimulation	GPR4	*GPR4*	*GPR19*	GPCR-Class A orphans
GPR132	Protons	Protons	Gα13 and Gαs family	Phospholipase C activation	GPR132	*GPR132*	*G2A*	GPCR-Class A orphans
Succinate receptor	Dicarboxylic acids	succinic acid, maleic acid	Gi/Go family Gq/G11 family	Adenylate cyclase inhibition Phospholipase C stimulation	GPR91	*SUCNR1*	*GPR91*	Succinate receptor

**Table 2 cells-09-02345-t002:** Differential expression of metabolite sensing-GPCRs in several types of cancer. This table indicates the neoplasias where an up-regulated or a down-regulated GPCR expression has been detected in human tumoral samples compared with the corresponding healthy tissues. (FFA, free fatty acid; HCA, hydroxycarboxylic acid; TA, trace amines; CaSR, calcium sensing receptor; TAAR1, trace amine associated receptor 1).

GPCR		DISREGULATION IN CANCER	
		DOWN-REGULATION	UP-REGULATION
**Fatty Acid Receptors**	**GPR43/FFA2**	Colon, Lymph Node, Metastatic Adenocarcinomas	Colon, Stomach
	**GPR41/FFA3**		
	**GPR40/FFA1**		Thyroid, Pancreas, Colon
	**GPR120/FFA4**		Colon, Glioma, Lung, Stomach, Pancreas, Kidney
	**GPR84**		Glioma, Lung, Stomach, Ovarian, Breast
	**GPR119**		Pancreas
**Hydroxycarboxylic Acid Receptors**	**GPR81/HCA_1_**		Pancreas, Breast
	**GPR109A/HCA_2_**	Colon, Breast, Skin	
	**GPR109B/HCA_3_**	Skin	
**Amino Acid and Related Metabolites Receptors**	**CasR**	Colorectal, Neuroblastoma	Pancreas, Breast, Prostate, Kidney, Lung
	**TAAR1**	Pancreas, Prostate, Liver	Stomach, Lung, Cervical, Esophagus
	**GPR35**	Testicular, Thyroid, Prostate	Stomach, Pancreas, Lung, Colorectal, Liver, Kidney, Endometrial
	**GPR142**		Pancreas, Stomach
**Bile Acid Sensing Receptor**	**TGR5**		Esophagus, Breast, Pancreas, Stomach, Colorectal, Liver, Testicular, Urothelial, Kidney
**pH-Sensitive Receptors**	**GPR4**	Cervical, Lung, Kidney	Breast, Ovarian, Colon, Liver, Kidney, Cholangiocarcinoma, Head and Neck, Renal
	**GPR65**	Hematological	Kidney, Ovarian, Colon, Breast, Metastatic Melanoma
	**GPR68**	Prostate, Kidney, Thyroid, Stomach, Esophagus	Pancreas, Cervical, Bladder, Breast, Ovarian, Testicular, Colon, Lung, Medulloblastoma, Head and Neck, Endo/Neuroendocrine, Thyroid, Stomach, Esophagus
	**GPR132**		Lung, Pancreas, Thyroid, Cervical, Endometrial, Breast, Testis, Kidney
**Citric Acid Cycle Intermediates Receptors**	**GPR91**		Lung

**Table 3 cells-09-02345-t003:** Functional studies performed describing the role of the Free Fatty Acid Receptors (FFAR) in several types of cancers. This table includes the literature which demonstrates the anti-cancerous and pro-cancerous role of each Fatty Acid Receptor in both in vitro cells or in vivo with mice.

GPCR	anti-CANCEROUS EFFECTS		pro-CANCEROUS EFFECTS	
	CANCER TYPE	MODEL	CANCER TYPE	MODEL
**GPR43/FFA2**	Colon	Overexpression of GPR43 reduced cell proliferation and increased apoptosis in colon cancer cells		
**GPR41/FFA3**	Liver	FFA3 mediates apoptosis induced by propionate and cisplatin in HepG2 cells	Ovarian	Over-expression of FFAR3 in CHO cells prevented the anti-proliferative and pro-apoptotic properties of butyrate
	Gastric	Activation of FFAR3 by SCFAs inhibit bovine epithelial cells proliferation		
**GPR40/FFA1**	Osteosarcoma	FFA1 activation with the agonist GW9508 stimulated the cell motile activity of MG63-R7 cells	Lung	FFA1 blockade with GW1100 reduced cell motile activities of RLCNR, LL/2 and A549 cells
			Ovarian	FFA1 activation promotes the proliferation of epithelial ovarian cancer (EOC) cells
			Prostate	FFA1 activation with oleic acid induced the proliferation and resistance to cytotoxic agents in Pancreatic cells PC3 and DU-145
	Fibrosarcoma	GPR40 knockdown in HT1080 cells enhanced cell motility and invasive activities	Breast	FFA1 activation with oleate induced the proliferation of MCF-7 cells
				FFA1 blockade with GW1100 reduced the cell growth rate of MCF cells treated with tamoxifen
			Melanoma	FFA1 stimulated cell motile activity of melanoma cells (A375 and G361 cells) treated with 12-O-Tetradecanoylphorbol-13-acetate
	Pancreas	GPR40 knockdown in PANC-1 cells showed increased cell motility		
**GPR120/FFA4**	Prostate	FFA4 activation with the agonist TUG-891 exerts inhibitory effects on LPA- and epidermal growth factor-induced proliferation and migration in DU145 and PC-3 cells	Pancreas	GPR120 knock-down reduced the low cell motility of PANC-1 cells
			Colon	FFA4 activation with GW9508 enhanced cell migration and induced a pro-angiogenic response in human colorectal carcinoma (CRC) cells
	Lung	GPR120 negatively regulated cellular functions during tumor progression in lung cancer RLCNR, LL/2 and A549 cells		Pharmacological activation of FFA4 promotes tumor growth in xenograft experiments in nude GPR120-KO mice
			Breast	FFA4 activation reduces the sensitivity of breast cancer cells (MCF-7 and MDA-MB-231) to epirubicin
	Melanoma	GPR120 knock-down increased the cell motile activity of A375 cells treated with TPA		Mice bearing MCF-7/ADM xenografts treated with AH7614 or GPR120-siRNA presented a reduced the tumor size and weight

**GPR84**			Leukemia	GPR84 blockade results in anti-inflammatory effects and antiproliferative effects in leukemic stem cells (LSCs)
**GPR119**				

**Table 4 cells-09-02345-t004:** Functional studies performed describing the role of the hydroxycarboxylic acid receptors (HCAR) in several types of cancers. This table includes the literature which demonstrates the anti-cancerous and pro-cancerous role of each hydroxycarboxylic acid receptor in both in vitro cells or in vivo with mice.

GPCR	anti-CANCEROUS EFFECTS		pro-CANCEROUS EFFECTS	
	CANCER TYPE	MODEL	CANCER TYPE	MODEL
**GPR81/** **HCAR1**			Breast	GPR81 knock-down reduced the proliferation rate and promoted the apoptosis of breast cancer cells MCF7
				Reduction of tumor growth and metastatic potential induced by the orthotropic xenotransplantation of MCF7-shGPR81 cells to nude mice
			Pancreas	GPR81 knock-down induced a rapid death of pancreatic carcinoma cells cultured in conditions of low glucose supplemented with lactate of
				Reduction of tumor growth induced by orthotropic xenotransplantation of Capan·II-shGPR81 cells to nude mice
**GPR109A/** **HCAR2**	Breast	GPR109A activation with butyrate induced apoptosis of breast carcinoma cells	Brain	GPR109A activation with butyrate increased the proliferation of glioblastoma cells
		Deletion of GPR109A increased tumor incidence of spontaneous breast cancer		
	Colon	Re-expression of GPR109A in colon cancer cells induced apoptosis, but only in the presence of its ligands butyrate and nicotinate		
		GPR109A activation with butyrate induced apoptosis of colon carcinoma cells		
		GPR109 activation with butyrate and niacin suppressed colonic inflammation and carcinogenesis		
**GPR109B/** **HCAR3**				

**Table 5 cells-09-02345-t005:** Functional studies performed describing the role of the amino acid, amino acid-related metabolites and bile acid sensing receptors in several types of cancers. This table includes the literature which demonstrates the anti-cancerous and pro-cancerous role of each amino acid, amino acid-related metabolites and bile acid sensing receptor in both in vitro cells or in vivo with mice. (CasR, calcium sensing receptor; TAAR1, trace amine associated receptor 1).

GPCR	anti-CANCEROUS EFFECTS		pro-CANCEROUS EFFECTS	
	CANCER TYPE	MODEL	CANCER TYPE	MODEL
**CasR**	Colorectal Carcinoma	The intestinal epithelium conditional CasR-KO mice presented intestinal hyperproliferation and were more susceptible to azoxymethane	Breast Cancer	CasR activation with calcium stimulated the proliferation of breast cancer cells (MDA-MB-231, T47D and MCF7 cells)
				CaSR-WT overexpression enhances MDA-MB-231 osteolytic potential of intratibially injected breast cancer cells in BALB/c nude mice
	Neuroblastoma	Overexpression of CasR reduced the proliferation and activated the apoptosis of neuroblastoma cell lines	Renal Cell Carcinoma	CasR increased cell migration and proliferation of human 768-O renal carcinoma cells
				Intracardiac injection of CaSR-transfected the RCC cells 768-O cells increased bone metastasis
			Prostate Cancer	CasR blockade with the antagonist calcilytics reduced the proliferation and migration of human prostate cancer cells
			Gastric Cancer	CaSR knockdown attenuated the CaCl2-enhanced migration and invasion of GES-1 and MKN45 cells
				CasR blockade with NPS-2143 antagonist reduced the proliferation, migration and invasion and promotes the apoptosis of AGS cells
**TAAR1**	Leukemia	TAAR1 activation with the agonists T1AM, o-PIT and tyramine induced the apoptosis of L3055 Burkitt lymphoma cells		
	Breast Cancer	Stimulation of TAAR1 with cadaverine exerted a beneficial effect against development of breast cancer		
**GPR35**	Colorectal Carcinoma	GPR35-KO mice prevented the inflammation-associated and spontaneous intestinal tumorigenesis	Non-Small-Cell Lung Cancer	GPR35 overexpression conferred drug resistance in non-small-cell lung A549 cells
**GPR142**				
**TGR5**			Gastric Cancer	TGR5 activation with the agonist deoxycholic acid activated ERK1/2, MAPK and EGF-R and its knock-down promoted the apoptosis of AGS cells
			Cholangiocarcinoma	TGR5 activation with the agonist INT-777 increased the proliferation, migration and mitochondrial metabolism of CCA cells
			Non-Small Cell Lung Cancer	TGR5 knockdown reduced the proliferation of H1975 and H1299 cells
				In xenograft tumor modes using H1975 NSCLC cells, mice with TGR5-shRNA cells showed a reduced relative tumor volume and a lower tumor weight

**Table 6 cells-09-02345-t006:** Functional studies performed describing the role of pH-sensitive and citric acid cycle intermediates-sensitive receptors in several types of cancers. This table includes the literature which demonstrates the anti-cancerous and pro-cancerous role of each pH-Sensitive and citric acid cycle intermediates-sensitive receptor in both in vitro cells or in vivo with mice.

GPCR	anti-CANCEROUS EFFECTS		pro-CANCEROUS EFFECTS	
	CANCER TYPE	MODEL	CANCER TYPE	MODEL
**GPR4**	Melanoma	GPR4 overexpression in melanoma cells reduced their migratory ability	Breast	GPR4-KO mice presented a reduced tumorigenesis after the orthotropic transplantation of cells from breast cancer cells
			Colon	GPR4-KO mice presented a reduced tumorigenesis after the orthotropic transplantation of cells from colon cancer cells
				Suppression of GPR4 in human cells attenuated tumor growth of subcutaneous xenografts
**GPR65**	Lymphoid	GPR65 overexpression increased the apoptosis of murine lymphoma cell line (WEHI7.2)	Breast	GPR65 overexpression in mammary epithelial NMuMG cells augmented their carcinogenic potential in vivo
		GPR65 silencing reduced the expression of the oncogene c-Myc and the proliferative activity of human lymphoma cells (U937)	Lung	GPR65 overexpression in LLC/GPR65 increased their carcinogenic potential in vivo in C57BL/6 mice
		GPR65-transfected cells showed a reduced tumorigenic and metastatic potential in SCID mice		
				GPR65 silencing reduced tumor development in nude mice
			Lymphoid	GPR65 overexpression in Lymphocytic Leukemia (CLL) cells increased their carcinogenic potential in vivo
**GPR68**	Prostate	GPR68 overexpression in PC3 cells reduced the metastasis after their orthotropic transplantation in nude mice	Prostate	GPR68-KO mice exhibited a reduced size of tumors induced by xenotransplantation of cells from prostate cancer TRAMP-C2 or RM-9 cells
	Breast	GPR68 transfection reduced the migratory ability of MCF7 breast cancer cells	Colon	GPR68 in fibroblasts promotes the formation of tumor spheroids in human colorectal carcinoma cells (HCT116)
				Reduced tumorigenic response in GPR68-KO mice injected with colon cancer MC-38 cells
	Myeloid	GPR68 activation with lenalidomide mediated the apoptotic effect of this compound in MDSL cells	Breast	GPR68 expression in human metastatic breast carcinoma MDA-MB-453 cells increased their tumorigenic behavior after the xenotransplantation in nude mice
			Melanoma	GPR68-KO mice exhibited a reduced size of tumors induced by xenotransplantation of cells from murine melanoma B16-F10 cells
**GPR132**	Leukemia	GPR132 deficiency promoted leukemogenesis in mice receiving BCR-ABL transduced bone marrow cells	Breast	GPR132 reduced the anticancerous effect of the PPARγ agonist rosiglitazone in an orthotropic transplantation model of breast cancer
		Pharmacological stimulation of GPR132 showed inhibitory effects in a murine model of acute myeloid leukemia		GPR132 favored a protumoral phenotype in tumor-associated macrophages and promoted lung metastasis in a murine model of breast cancer
**GPR91**			Lung	GPR91 expression in lung tumor A549 cells favored lung metastasis in mice after their xenotransplantation in nude mice
			Colon	GPR91 activation by succinate increased the expression of EMT genes in HT29 cells

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
