# Peer review of "Metabolite Sensing GPCRs: Promising Therapeutic Targets for Cancer Treatment?"

_cells, 2020, doi:10.3390/cells9112345_

Round 1
Reviewer 1 Report
The present manuscript is a review article which reports the putative role of the metabolite sensing G protein-coupled receptor as therapeutic targets for cancer treatment. This review is clear, detailed and very informative demonstrating, the important role of GPCRs in cancers. The review reveals that some metabolite sensing receptors are over- and/or under-expressed in various cancers in which it impacted the proliferation and/or apoptosis in cancer cells. However, some minor points could be added to improve the manuscript:
- In tables 3 to 6, the authors display the functional studies of various GPCRs in terms of anti- and pro-cancerous effects. However, the authors only report the “Cancer Type” and “Model”. The indication of the main action of these different GPCRs on cancer should be indicated to help the reader.
- As mentioned along this review, the role of these GPCRs in cancer is sometimes controversial in terms of expression and/or direct effects such as proliferation or apoptosis…. It may be important to discuss this in the conclusion paragraph.
Author Response
REFEREE 1
The present manuscript is a review article which reports the putative role of the metabolite sensing G protein-coupled receptor as therapeutic targets for cancer treatment. This review is clear, detailed and very informative demonstrating, the important role of GPCRs in cancers. The review reveals that some metabolite sensing receptors are over- and/or under-expressed in various cancers in which it impacted the proliferation and/or apoptosis in cancer cells. However, some minor points could be added to improve the manuscript:
In tables 3 to 6, the authors display the functional studies of various GPCRs in terms of anti- and pro-cancerous effects. However, the authors only report the “Cancer Type” and “Model”. The indication of the main action of these different GPCRs on cancer should be indicated to help the reader.
We appreciate so much the suggestion of the referee and now we have included in Tables 3 to 6 the main effects attributed to each GPCR in each cancer type.
As mentioned along this review, the role of these GPCRs in cancer is sometimes controversial in terms of expression and/or direct effects such as proliferation or apoptosis…. It may be important to discuss this in the conclusion paragraph.
We have modified the conclusion paragraph following the referee’s suggestion.

Reviewer 2 Report
This is a very comprehensive review that is well-organized and clear on an important topic. I have a few minor comments:
1) In several instances, including for example the description of FFAR1, specific GPCRs are implicated as tumor suppressors and oncogenes, depending on the cellular context. Some attempt should be made to explain these contradictory findings on a molecular level, if possible.
2) I do not believe the description of amino acids on line 335 is accurate.
Author Response
REFEREE 2
This is a very comprehensive review that is well-organized and clear on an important topic. I have a few minor comments:
1) In several instances, including for example the description of FFAR1, specific GPCRs are implicated as tumor suppressors and oncogenes, depending on the cellular context. Some attempt should be made to explain these contradictory findings on a molecular level, if possible.
As pointed out by the referee, many of the receptors analyzed show anti- and pro-cancerous effects in different studies. In general, the opposite roles of a particular GPCR in cancer have been characterized in different kinds of tumors. We have postulated potential reasons for these apparently contradictory evidences when found (e.g. the kind of cells expressing GPR65 inside the tumor) but this has been anecdotic. In most cases, it is difficult to infer a global explanation for the results published in independent investigations, and this situation clearly justifies the necessity of new comparative studies specially designed to answer these questions.
2) I do not believe the description of amino acids on line 335 is accurate.
According to the referee´s suggestion, the description of amino acids has been changed.
